# Enhancing the “Broaden and Build” Cycle of Attachment Security in Adulthood: From the Laboratory to Relational Contexts and Societal Systems

**DOI:** 10.3390/ijerph17062054

**Published:** 2020-03-20

**Authors:** Mario Mikulincer, Phillip R. Shaver

**Affiliations:** 1Baruch Ivcher School of Psychology, Interdisciplinary Center (IDC) Herzliya, HaUniversita 8 Street, Herzliya 46150, Israel; 2Department of Psychology, University of California, Davis, One Shields Avenue, Davis, CA 95616-8686, USA; prshaver@ucdavis.edu

**Keywords:** attachment security, security priming, close relationships, well-being, adult development

## Abstract

Attachment theory emphasizes both the importance of the availability of caring, supportive relationship partners, beginning in infancy, for developing a sense of safety and security, and the beneficial effects of this sense of security on psychosocial functioning and physical and mental health. In this article, we briefly review basic concepts of attachment theory, focusing on the core construct of attachment security and present evidence concerning the ways in which this sense can be enhanced in adulthood. Specifically, we review findings from laboratory experiments that have momentarily enhanced the sense of attachment security and examined its effects on emotion regulation, psychological functioning, and prosocial behavior. We then review empirical findings and ideas concerning security enhancement by actual relationship partners, non-human symbolic figures, and societal systems in a wide variety of life domains, such as marital relationships, psychotherapy, education, health and medicine, leadership and management, group interactions, religion, law, and government.

## 1. Introduction

In his original and seminal exposition of attachment theory, Bowlby [1] emphasized the importance of positive interactions with caring, loving relationship partners and the resulting sense of attachment security (confidence that one is worthy and lovable and that others will be supportive when needed). Bowlby argued that this sense of security is important for personal adjustment, psychosocial functioning, and physical and mental health. It allows people to cope effectively with threats, maintain self-esteem and emotional equanimity, form harmonious and satisfying close relationships, broaden skills and perspectives, and fully enjoy life [2] In this article, we focus on the “broaden and build” cycle of attachment security [3] and present evidence on the ways in which this cycle can be enhanced in adulthood. Specifically, we review findings from laboratory experiments that have momentarily enhanced the sense of attachment security and revealed the effects on emotion regulation, psychological functioning, and prosocial behavior. We then review findings concerning the enhancement of the sense of attachment security by actual relationship partners, non-human figures, and societal systems and the development of security-enhancement interventions in a wide variety of domains, such as marital relationships, psychotherapy, education, medicine, leadership and management, law, and government. 

## 2. Attachment Theory: Basic Concepts

According to Bowlby [1], human beings are born with an innate psychobiological system (the *attachment behavioral system*) that motivates us to seek proximity to supportive others (*attachment figures*) in times of need as a way to reduce anxiety and obtain protection. According to Bowlby [4], attachment figures are expected to function as a *safe haven* in times of need—providing protection, comfort, and relief—and as a *secure base*, encouraging the autonomous pursuit of non-attachment goals while remaining available if needed. In this way, attachment figures provide a sense of attachment security, instill feelings of being loved and cared for, and thereby facilitate effective functioning and thriving in non-attachment activities, such as exploration, learning, interpersonal exchanges, and sexual mating. Although the attachment system is most important early in life (when it is a matter of life or death), Bowlby [4] believed it is active across the life span, is manifest in thoughts and behaviors related to support-seeking in times of need, and produces feelings of being loved and cared for, which continues to be beneficial for psychosocial functioning and health. This claim provided the impetus for subsequent theorists and researchers to conceptualize and study attachment-related processes in adulthood and examine ways to enhance the sense of attachment security in adults, e.g., [5,6,7].

Bowlby [8] also described important individual differences in the extent to which a person holds an underlying sense of security. In his view, this sense is rooted in actual relationships with attachment figures who provide a safe haven and secure base. These actual relationships form the basis of what Bowlby called *internal working models* (scripts or schemas) of self and others. However, when attachment figures have not been sensitive and responsive to one’s bids for proximity and support, one’s sense of personal safety, security, and lovability is shaky or nonexistent, and others’ benevolence and reliable supportiveness are in doubt. These frustrating and emotionally painful experiences result in what we call insecure *attachment orientations* [2], which can be conceptualized as regions in a continuous two-dimensional space [9,10]. One dimension, attachment-related *avoidance*, reflects the extent to which a person distrusts others’ benevolence and defensively strives to maintain independence and emotional distance from relationship partners. The other dimension, attachment anxiety, reflects the extent to which a person worries that others will not be responsive in times of need, which increases the tendency to be preoccupied with, and intrusive in, close relationships.

One should take into account that a person’s habitual attachment orientation in close relationships can be conceptualized as the top node in a complex network of attachment working models. Some of the non-dominant representations apply only to specific people and relationships and others apply only in certain relational contexts [11,12,13,14]. These more specific working models can be activated by actual or imagined interactions with attachment figures, even if they are not congruent with a person’s habitually dominant attachment orientation [2]. These activated mental representations can shift one’s context-specific or partner-specific working models and induce some malleability in the top-node (usually dominant) attachment orientation. Therefore, repeated activation of a specific working model may eventually alter one’s dominant attachment orientation [2]. In this article, we focus on the psychological effects of activating mental representations of attachment security in the laboratory and real-life settings.

## 3. The Broaden-and-Build Cycle of Attachment Security

The enhancement of the sense of attachment security resulting from actual or imagined interactions with sensitive and responsive attachment figures promotes what we, following Fredrickson [15], call a “broaden and build” cycle of attachment security, which builds one’s resilience and broadens one’s perspectives and skills [2]. Security enhancement fosters both positive working models of others and the self (others are benevolent and trustworthy, and the self is valuable and lovable thanks to having been valued and loved by others) and what Waters and Waters [13] called a “secure-base script,” which sustain distress management, optimism, and hope while dealing with life’s inevitable adversities and challenges. The following if-then propositions are the core components of this script: “If I encounter an obstacle and/or become distressed, I can approach an attachment figure for safe-haven support; he or she is likely to be responsive; I will experience relief and comfort; I can then return to non-attachment activities with the confidence that support will be available when needed”. Research has shown that people who are more secure with respect to attachment hold in mind richer secure-base scripts when narrating threat-related stories or dreams [14] and score higher on measures of interpersonal trust, self-esteem, and optimism [2]. 

In our view, both interactions with responsive attachment figures (at the interpersonal level) and the activated positive working models and secure-base script (at the intrapersonal level) contribute to one’s emotional strength and composure in the face of adversities [2]. Moreover, they facilitate the formation and maintenance of satisfying close relationships [16] and allow people to fully engage in non-attachment activities (e.g., exploration, play, learning) rather than devoting self-regulation resources to attachment-related worries and defenses. Moreover, holding confident expectations of secure-base support, people can take calculated risks, experiment and make mistakes, and engage in activities that challenge their knowledge and perspectives. Several studies have shown that people who are more secure with respect to attachment are more likely to thrive in non-attachment activities, such as learning, caregiving, and sex [17,18,19].

Theoretically, the broaden-and-build cycle of security is renewed every time a person interacts with a loving, security-enhancing attachment figure in times of need [2]. During infancy, these interactions contribute to the emergence of the broaden-and-build cycle of security. In later stages of development, interactions with a loving figure reinforce the existing cycle in secure people. More important from a clinical or interventionist perspective, experiencing such interactions, or remembering or imagining them, may activate security-related representations even in insecure people. That is, security-enhancing interactions, especially when they are reliably repeated over time, may allow insecure people to feel loved, appreciated, and cared for; initiate a broaden-and-build cycle of security; and gradually feel more fundamentally secure. In adulthood, the broaden-and-build cycle of security can be renewed by reminders of loving relationship partners or groups and imaginary interactions with them and with various non-human or symbolic figures (e.g., pets, God) or societal systems (e.g., law, government) that help people to feel supported and safe. In our view, these actual or imaginary experiences support a person’s maintenance of a secure-base script and thereby foster relaxed openness to current experience, counteract pessimistic and self-defeating beliefs, and free self-regulatory resources to engage productively in non-attachment activities. In both the short and long run, these processes are likely to have positive effects on functioning and health. 

In the following sections, we review findings from laboratory experiments concerning the psychological effects of contextually boosting adults’ sense of attachment security. We then review findings from studies concerning the psychological effects of actual interactions with a responsive other in marital relationships and psychotherapeutic, educational, medical, and organizational settings. Finally, we review findings on the potential security-enhancing effects of responsive non-human figures and societal systems. While providing these reviews, we also show how the reviewed findings might be applied to developing effective interventions aimed at enhancing security and improving psychological functioning and health. 

## 4. Enhancing Security in the Laboratory

Attachment researchers have used well-validated experimental techniques to activate mental representations of security within the laboratory—a process we call *security priming* [20]—and have examined the immediate psychological effects of this activation. These techniques include explicit and implicit exposure to pictures suggesting attachment security (e.g., a Picasso drawing of a mother warmly cradling an infant in her arms) or security-related words (e.g., hug, love, safe), and guided imagery concerning the receipt of safe-haven support from an attachment figure in times of need. Beyond these generic manipulations, studies have also used more person-tailored, idiosyncratic priming techniques, such as explicit and implicit exposure to names of actual people nominated by participants as security providers; mental visualization of the face of a security provider; and viewing a photograph of this person. In some of these studies, participants received the 6-item WHOTO scale [21] and provided the name of a specific person who fits the role of a security provider as described in each of the items (proximity seeking, safe haven, secure base). In other studies, participants received instructions to think about a specific person who supports and comforts them when they are distressed and whom they trust will be available and responsive when needed [2]. The common denominator of all these person-specific techniques is that the primed figure accomplishes both safe-haven and secure-base functions. The effects of these primes have been compared with the effects of emotionally positive but attachment-unrelated stimuli or emotionally neutral stimuli [2].

Findings indicate that security priming enhances interpersonal trust, self-esteem, and pro-relational expectations [22,23]. Moreover, it improves mood and facilitates emotion regulation. For example, Mikulincer, Hirschberger, Nachmias, and Gillath [24] found that implicit exposure to the names of participants’ security providers (nominated in the WHOTO questionnaire), compared with the names of close others or acquaintances who were not nominated as attachment figures, improved implicit mood (greater liking of previously unfamiliar Chinese ideographs) even in a threatening context. Subsequent studies showed that security priming accelerated emotional recovery after recalling an upsetting event [25], inhibited unwanted intrusions of distressing memories [26], attenuated activation in brain areas implicated in reactivity to social threats [27,28], and increased physiological signs of relaxation during stress exposure [29]. There is also evidence that security priming reduces anxiety and depression in clinical and non-clinical samples, enhances mindful attention, and causes distressed people to become more open-minded about therapy [30,31,32,33].

Laboratory experiments also show that security priming facilitates fuller engagement in exploration and learning. For example, Green and Campbell [34] asked people to read generic sentences describing secure or insecure close relationships and found that a secure prime, compared with an insecure prime, led to greater endorsement of exploration-related behavior and greater liking for novel pictures. Moreover, Luke, Sedikides, and Carnelley [35] found that security priming, as compared with neutral priming, increased both vitality and willingness to learn and that these effects could not be explained by positive effect. Additionally, Mikulincer, Shaver, and Rom [36] found that implicit exposure to the name of a security provider (nominated in the WHOTO) led to better performance on a creative problem-solving task than implicit exposure to the names of others who were not nominated as attachment figures. 

The broadening effects of a laboratory security enhancement are also evident in prosocial feelings and behaviors. For example, Mikulincer, Shaver, Gillath, and Nitzberg [37] found that, as compared with neutral priming, both implicit and explicit forms of security priming (implicit exposure to the name of a security provider nominated in the WHOTO, asking participants to think about this figure) increased feelings of compassion and actual helping behavior toward a stranger who was becoming increasingly distressed while performing a series of aversive tasks. More recently, security priming was found to foster effective care for a romantic partner who was disclosing a personal problem or discussing personal goals [38,39]. Dating couples came to a laboratory and were video-recorded during an interaction in which one of them (“the care-seeker”) disclosed a personal problem or future goals to the other (“the caregiver”). Prior to this interaction, caregivers were exposed to either the names of security providers (nominated in the WHOTO) or the names of acquaintances. Findings indicated that security priming, as compared to neutral priming, increased actual supportive behavior (as coded by independent judges) toward the disclosing partner.

Overall, both implicit and explicit priming of mental representations of attachment security tend to have positive psychological consequences. However, Mikulincer and Shaver [2] concluded that the psychological effects of explicit security priming tend to be more dependent on a participant’s dispositional attachment orientation than the effects of implicit security priming, perhaps because explicit (conscious) primes may activate certain kinds of defenses that are not activated by implicit primes. That is, some of the reviewed studies have shown that individuals who are more chronically attachment-avoidant or -anxious tend to respond to explicit security primes differently from more secure individuals, e.g., [26,36]. Unfortunately, the current state of the literature does not allow us to delineate the specific explicit stimuli and manipulations that are the most effective security primes for enhancing security in either attachment-avoidant or attachment-anxious people. This is a major task for future experimental work on security priming.

The laboratory security priming techniques are robust, replicable, reliable, and distinct from positive mood inductions [40], but their effects disappear as participants finish the experiment and are difficult to generalize to real-life settings. Although there is initial evidence that repeatedly exposing participants to security primes over several occasions (spanning from one week to several weeks) results in positive effects that are sustained for days [22,31,41], more research is needed before drawing conclusions about the long-term impact of repeated security priming in real life. One step in this direction would be to design electronic applications (apps) that deliver diagrammatic security primes (e.g., photo, voice, or video of a security provider) or ask people to engage in a brief security-enhancing exercise (e.g., writing about a security provider or a specific supportive interaction with him or her) each time the app is activated. A more advanced step would be to construct a sophisticated machine-learning app that could prime security in the ways described above immediately upon detection of elevations in a person’s physiological or behavioral signs of arousal and distress. (This kind of detection is already possible with some hi-tech wristwatches.) In this way, the app could boost security in the same way security primes have been shown to do in the lab or the way a supportive attachment figure does in normal social life. Researchers could then determine whether using these devices over an extended time period has lasting effects on functioning and health. 

## 5. Enhancing Security in Real-Life Social-Relational Contexts

Attachment researchers have also examined the effects of enhancing security within actual relational contexts, where a real relationship partner can provide a safe haven and secure base when a person feels a need for this. These contexts include ones with close partners (e.g., family members, close friends, dating partners, spouses) as well as ones in which the potential security provider is a domain expert and occupies the role of a “stronger and wiser” (Bowlby’s terms) caregiver in a formal role hierarchy (e.g., teacher, coach, therapist, manager). In egalitarian relationships, such as friendships and romantic relationships, each partner can informally occupy the role of a “stronger and wiser” caregiver when the other partner is in need and asks for safe-haven or secure-base support. However, when one partner chronically occupies the role of the needy care-seeker or monopolizes the role of the stronger and wiser caregiver in these supposedly egalitarian relations, it unbalances and potentially damages relationship quality and stability [2].

At the beginning of these hierarchical or egalitarian relationships, people’s dominant attachment orientation (either secure or insecure), especially the associated model of others, can be automatically projected onto the potential security provider at the beginning of the relationship, thereby preventing any change in the working models. However, this figure’s sensitivity and responsiveness to bids for proximity and support might counteract this projection and enhance attachment security, even among insecure people. This reasoning has been the basis for developing attachment-based interventions for various kinds of relational situations. These interventions target the potential security provider as an agent of change and attempt to heighten his or her responsiveness and capacity to provide empathic and effective care to enhance others’ autonomous growth and thriving.

This reasoning was first examined in the domain of parent–child relations. Numerous cross-sectional and prospective longitudinal studies found that parents’ responsiveness to their infants’ signals and needs contributed to the children’s security in relation to the parents (in Ainsworth’s Strange Situation; [9]) and more favorable developmental outcomes [42]. There is also extensive evidence that parents’ attachment orientations contribute to their child’s attachment security and favorable psychological development [43]. Longitudinal studies have revealed that the beneficial effects of parents’ responsiveness during infancy tend to persist over time and contribute to adolescent and adult well-being and functioning [44].

Based on these findings, child psychologists have created attachment-based intervention programs aimed at heightening parents’ responsiveness as a means of enhancing children’s security and their positive development. Some of these programs include short-term interventions (5–16 weeks), mostly relying on parents’ psycho-education and video feedback of their behavior during interactions with their infants. Research findings clearly indicate that infants’ attachment security is enhanced when parents participate in these short-term programs, especially when parents themselves show improved post-intervention responsiveness [45]. Similar positive effects have been obtained in studies of more intensive and longer (20 weeks to 1 year) intervention programs [46,47,48]. Most of these interventions include psychotherapy aimed at correcting parents’ attachment-related fears and defenses that interfere with care provision.

## 6. Romantic and Marital Relationships

In adulthood, a romantic or marital partner is often a person’s primary attachment figure [49]. Therefore, attachment research has examined security enhancement within romantic relationships, and findings are being applied to improve the effectiveness of couple and marital counseling. During the past 40 years, hundreds of studies have documented the crucial contribution of a person’s dominant attachment orientation to motives, cognitions, feelings, and behavior in the context of couple and marital relationships [2,16]. At the same time, there is growing evidence that supportive and loving couple interactions attenuate partners’ distress and contribute to psychological well-being, physical health, and longevity [50,51]. Correlational studies have also indicated that actual interactions with a responsive dating partner or spouse promote a wide variety of pro-relational cognitions and behaviors that heighten relationship stability and satisfaction [52]. There is also correlational evidence that the availability and responsiveness of a romantic partner in times of need is associated with the other partner’s within-relationship sense of security and heightened feelings of relationship satisfaction [53]. 

On the basis of such research, Arriaga, Kumashiro, Simpson, and Overall [54] proposed the *Attachment Security Enhancement Model* (ASEM), in which they claim that one partner’s sensitive and responsive behaviors can buffer or reduce the other partner’s attachment anxiety or avoidance during moments of relational tension, and can thereby foster attachment security within a relationship over the long run. Indeed, Arriaga, Kumashiro, Finkel, VanderDrift, and Luchies [55] found in a longitudinal study of newlywed couples that perceptions of partners as available and responsive were associated with reductions in attachment avoidance 12 months later, and that perceptions of partners as accepting and valuing one’s needs and goals predicted subsequent reductions in attachment anxiety. In addition, Stanton, Campbell, and Pink [56] reported that engagement in intimacy-promoting activities with a romantic partner led to reductions in attachment-related avoidance one month later. The beneficial effects of a partner’s responsiveness on attachment insecurities were also evident in Lavi’s [57] 8-month longitudinal study with newly committed couples. Specifically, within-relationship attachment anxiety and avoidance decreased during the 8-month study period as a direct function of a partner’s actual sensitive and responsive behaviors in dyadic interactions (as coded by external judges) at the beginning of the study.

The security-enhancing role of a romantic partner’s responsiveness is also one of the core aspects of Sue Johnson’s [58] *Emotion-Focused Therapy* (EFT) for couples. Johnson [58] conceptualizes relationship distress as resulting from one partner’s lack of responsiveness to the other partner’s support-seeking bids and from their own unacknowledged and unmet attachment needs (*attachment injuries*). EFT helps partners acknowledge basic attachment needs, insecurities, and injuries and improve their ability to respond to each other with sensitive and responsive care, resulting in more positive and pro-relational interactions. There is growing evidence that heightening partners’ functioning as a secure base to one another within the context of EFT dramatically reduces relationship distress and improves relationship quality [59].

## 7. Counseling and Psychotherapy 

In his 1988 book, *A Secure Base*, Bowlby developed a model of therapeutic change focused on the ability of a responsive therapist to enhance clients’ security, and encourage them to explore and understand their painful attachment experiences, identify and revise insecure working models of self and others, and acquire more adaptive patterns of relating. In his view, clients typically enter therapy in a state of distress and psychological pain, which automatically activates their attachment system and causes them to yearn for support and relief. Attachment needs are easy to direct toward therapists, because therapists, at least when a client believes in their healing powers, are perceived as “stronger and wiser” caregivers. Therapists are expected to know better than their clients how to deal with the clients’ problems, and they occupy the dominant and caregiving role in the relationship. As a result, the therapist can easily become a potential provider of security and a target of the client’s projection of attachment-related worries and defenses. As a result, the therapist’s responsiveness to clients’ support-seeking bids becomes crucial in enhancing clients’ attachment security and fostering positive therapy outcomes. 

Research has provided support for Bowlby’s [4] conceptualization of psychotherapy. Numerous studies have shown that clients’ pre-therapy attachment orientations bias their attitudes toward therapists and therapy, shape the establishment of a good working alliance, and affect therapeutic outcomes [2,60,61]. In addition, there is evidence that clients tend to perceive therapists as security providers [62] and that therapists’ responsiveness has beneficial effects on therapy outcomes [63]. Studies have also found that the formation of clients’ secure attachment to a therapist has beneficial effects on therapeutic change [64]. There is also growing evidence that therapy can move clients away from insecure and toward secure attachment orientations, and that this movement is a good indication of effective treatment. For example, Travis, Bliwise, Binder, and Horne-Moyer [65] found an increase in clients’ reports of secure attachment across the course of time-limited dynamic psychotherapy, and this increase was associated with decreases in the severity of psychiatric symptoms. Similarly, Maxwell, Tasca, Ritchie, Balfour, and Bissada [66] found that attachment insecurities decreased during group psychotherapy and that this decrease predicted improvement in clients’ well-being and functioning up to 12 months after therapy.

Several evidence-based therapies have incorporated Bowlby’s [4] principles of therapeutic change in both individual and group psychotherapy. Among these therapies are the following: *Mentalization-Based Therapy* [67], *Accelerated Experiential-Dynamic Psychotherapy* [68], *Attachment-Based Group Psychotherapy* [69], and *Group Psychodynamic Interpersonal Psychotherapy* [70]. These attachment-based interventions explicitly recognize the positive therapeutic effects of interventions that focus on enhancing attachment security. Moreover, they highlight the importance of a sensitive and responsive therapist for enhancing security and revising maladaptive working models. There is growing evidence that these attachment-based approaches are more effective than other cognitive-behavioral or psychodynamic approaches in improving mental health among patients diagnosed with eating disorders, depression, and personality disorders [66,69,71].

## 8. Education

Security can be also enhanced in teacher–student relationships, with positive implications for academic performance and socio-emotional adjustment to school [72,73]. Theoretically, teachers, mainly at the kindergarten and elementary school levels, function as context-specific attachment figures who can provide comfort and support within the school setting. Moreover, they can function as a secure base from which children can explore and learn, take risks, and even make mistakes, with the confidence that their teacher’s support will be available when needed [74]. As a result, children whose teachers function as effective security providers can maintain an open and confident attitude toward learning and remain relaxed while dealing with school-related task and challenges. In support of this view, many studies have shown that elementary-school children whose teacher is warmer and more emotionally responsive tend to exhibit better socio-emotional and academic adjustment to school [75]. Moreover, field experiments have found that improving teachers’ responsiveness to students’ needs improves elementary-school children’s academic functioning and socioemotional adjustment [76,77]. 

Based on such research findings, Pianta, La Paro, and Hamre [78] developed a systematic classroom observation system that captures the extent to which a teacher is responsive to children’s support-seeking bids and provides a secure climate to explore and learn: the Classroom Assessment Scoring System (CLASS). The primary domains assessed in the CLASS are emotional support (teacher’s ability to manage students’ emotional needs), classroom organization (teacher’s ability to manage students’ behaviors), and instructional support (teacher’s ability to provide constructive and supportive feedback to students’ academic efforts and performance). The CLASS has been found to have good psychometric qualities and to predict students’ academic functioning and adjustment to school [79,80].

The CLASS has also been used to enhance a teacher’s functioning as a secure base and improve student–teacher interactions. For example, evidence-based professional development programs, such as My Teaching Partner (MTP) [81], use the CLASS framework to analyze videotaped teacher–student interactions and provide feedback to teachers on their functioning as a secure base. In the MTP program, for example, teachers watch videotaped teacher–student interactions of highly responsive teachers and work with a coach in identifying security-enhancing responses to students’ needs. The coach also provides ongoing constructive feedback on their own interactions with students and creates an action plan to change their interactions with students and improve their functioning as a secure base. Studies have found the MTP effective in enhancing teachers’ responsiveness and improving children’s academic functioning and school adjustment [80]. 

## 9. Health and Medicine

From an attachment perspective, physical pain, injuries, and illnesses can provoke fear and distress, which automatically activates the attachment system [82]. As a result, needs for protection and support and characteristic attachment orientations, including working models of self and others, are activated and directed toward people who can reduce illness-related worries and distress. According to Maunder and Hunter [82], this kind of attachment-system activation is likely to be directed toward physicians and other healthcare providers in medical settings, because they are perceived as a source of knowledge, healing, and physical safety. Thus, we can expect clients to project their attachment concerns and orientations onto their relationships with physicians, which may be relevant to explaining individual differences in the healing process. In addition, physicians’ responsiveness to clients’ support-seeking bids can enhance clients’ security and contribute to distress management, compliance with treatment, and the entire healing process.

Based on this reasoning, Maunder and Hunter [83] constructed a self-report scale to assess whether or not a healthcare provider functions as a safe haven (e.g., “In some circumstances, I might count on this person to help me feel better”) and a secure base (e.g., “This person makes me feel more confident about my health”). Patients were asked to nominate healthcare providers “who matter to you more than others” and to complete the scale for each of the identified providers. The scale showed adequate internal consistency, and 91% of the participants were able to identify at least one healthcare provider who mattered most, and the majority of them appraised these healthcare providers as fulfilling safe haven and secure base functions. Of course, this is only a preliminary study, and more data should be collected on the psychometric properties of this scale. 

Research also provides evidence that attachment orientations are relevant for explaining individual differences in health-related behaviors. For example, attachment anxiety and avoidance have been associated with less engagement in health-promoting behaviors, such as maintaining a healthy diet or engaging in physical activity, and more engagement in health-related risky behaviors such as smoking, drinking, drug abuse, and disordered eating or dieting [84,85]. There is also consistent evidence that attachment insecurities are associated with reduced adherence to medical regimens [86], more negative attitudes toward physicians and poorer trust in them [87], and slower restorative biological processes [88], which in turn counteract the healing process.

Despite the cumulative evidence highlighting the relevance of attachment theory for health and medicine, there is no systematic research program on the contribution of physicians’ responsiveness to clients’ health and physical recovery. In our review of the literature, we found only one study reporting that physicians’ attachment insecurities, which probably make them less responsive to clients, were associated with clients’ lower satisfaction with treatment [89]. Moreover, there is no evidence-based medical training program aimed at cultivating physicians’ functioning as a safe haven and secure base. However, in their 2015 book, *Love*, *Fear*, and *Health*, Maunder and Hunter provided practical recommendations to healthcare providers about how to manage clients’ attachment insecurities and how to make clients feel more secure. 

In addition, although not derived from attachment theory and research, several psychoeducation programs have been developed to enhance medical students’ and physicians’ capacity to respond empathetically to patients, e.g., [90,91]. However, most of these intervention evaluations suffer from poor research designs (e.g., non-random assignment, lack of a valid control condition) and inadequate assessment of long-term effects [92]. We are confident that construing these programs as means for enhancing patients’ attachment security and helping medical students and physicians to reduce patients’ insecurities would aid healing and reduce expenses for both clients and the medical system. 

## 10. Leadership and Management 

From an attachment perspective, there is a close correspondence between leaders (e.g., managers, political and religious authorities, supervisors, and military officers) and attachment figures. “Leaders, like parents, are figures whose role includes guiding, directing, taking charge, and taking care of others less powerful than they and whose fate is highly dependent on them” [93] (p. 42). That is, leaders often occupy the role of “stronger and wiser” caregivers and can provide a secure base for their subordinates [94]. Therefore, a responsive leader can enhance subordinates’ sense of security with positive implications for their self-esteem, competence, autonomy, and well-being. Moreover, a leader’s inability or unwillingness to respond sensitively and supportively to subordinates’ needs can magnify their anxieties and lead to feelings of demoralization and behavioral disengagement. In these cases, a non-responsive leader can transform what began with the promise of a secure base into a destructive, conflicted, hostile relationship that is damaging to subordinates and the organization to which they belong. 

In two studies conducted with Israeli combat soldiers and their direct officers, Davidovitz, Mikulincer, Shaver, Ijzak, and Popper [95] provided empirical support for the impact of a leader’s responsiveness on subordinates’ functioning and mental health. In one study, an officer’s ability to provide effective emotional and instrumental support to his soldiers in times of need (as rated by himself and his soldiers) was positively associated with his soldiers’ reports of socioemotional functioning and task performance in the military unit. In a second study, Davidovitz et al. [95] found that soldiers’ appraisal of their officer as a secure base during combat training (i.e., the officer’s ability and willingness to accept and care for his or her soldiers rather than rejecting and criticizing them) produced positive changes in soldiers’ mental health two and four months later. 

Subsequent studies have extended Davidovitz et al.’s [95] findings to business organizations, showing that managers’ responsiveness contributes positively to workers’ job satisfaction, organizational commitment, and psychological well-being [96,97,98]. These findings were conceptually replicated in studies examining relationships between school directors and teachers [99] and between coaches and athletes [100]. Using an experimental manipulation of supervisor behavior, Game [101] found that less secure workers reacted to a manager’s cold and rejecting behavior with greater distress. Future studies should extend this line of research and examine the potential effects of managers’ responsiveness to actual task performance. This is important because some organizations encourage high expectations and harsh feedback in the service of achieving peak performance. Such approaches would not necessarily lack safe-haven or secure-base provisions, but one should systematically examine whether managers’ responsiveness has positive effects on task performance within these competitive and demanding organizational settings. 

Although these findings support the beneficial effects of supervisors’ and managers’ provision of a safe haven and secure base for their subordinates, we know of no leadership development program based on attachment-theory principles. However, some attachment-theory principles can be found in positive leadership programs that train leaders to be emotionally available, mentor their subordinates, attend to and validate their subordinates’ needs, recognize their accomplishments, and encourage their autonomous growth [102]. In fact, organizational scientists and professionals are becoming more aware of attachment theory and the benefits of cultivating emotionally safe organizations and transforming managers into security-enhancing attachment figures.

## 11. Other Sources of Security Enhancement in Adulthood

During adolescence and adulthood, groups, institutions, and symbolic personages (e.g., God, the Buddha, or the Virgin Mary) can also be used as sources of a safe haven and secure base [2]. Therefore, confidence in the responsiveness of these symbolic figures might heighten positive working models of the self and others and improve psychosocial functioning and health even among people who are primarily insecure. However, unlike actual relationship partners whose sensitive and responsive behavior during dyadic interactions can counteract the projection of negative working models and provide insecure people with doses of security, these symbolic figures are a mere reflection of insecure people’s working models and not a separate entity whose autonomous responses can disconfirm dominant pessimistic expectations, worries, and doubts. That is, attachment to these symbolic figures is likely to result in a repetition of insecure people’s negative history of close relationships—a process that Kirkpatrick [103] explained in terms of a “correspondence” hypothesis when discussing attachment to God. As a result, the recruitment of potential symbolic attachment figures would probably fail to enhance security and to improve functioning and health unless people can generate more confident and trustworthy attitudes toward these figures and break the projection of dominant negative working models.

These ideas have received strong support in studies examining the *religion-as-attachment* model [104,105,106]. According to this model, God can serve as a protective attachment figure, and theistic believers’ relationship to God can be examined through the lens of attachment theory. First of all, believers tend to turn to God in times of need, and their prayers are often requests for assistance and protection. Indeed, studies of adult theistic believers have shown that exposure to threatening stimuli increases their wish to be close to God and makes God-related mental images more available [107,108]. Moreover, findings indicate that people who have insecure attachment orientations in close interpersonal relationships are more likely to approach religion and God in search of a compensatory safe haven during personal crises or illnesses [109,110,111]. However, there is also evidence that this compensation process may not be satisfying and that insecure attachment orientations in close relationships are projected onto God. Specifically, less secure people in close relationships are less likely to view God as a loving and caring figure and to feel secure in their attachment to God [112,113,114]. This may reduce the benefits they obtain from religion when it comes to emotion regulation, interpersonal functioning, and health [115,116]. 

Nevertheless, Davis, Granqvist, and Sharp [117] have raised the possibility that some insecure believers can “earn” a certain degree of attachment security by reparative experiences with religious leaders and members of their faith community or within the context of psycho-spiritual interventions aimed at strengthening secure attachment to God. There is accumulating evidence showing that participation in psycho-spiritual interventions, such as pastoral counseling or spiritually integrated psychotherapy [118,119], increase positive images of God and lead to positive changes in a person’s functioning and health. However, more systematic research is needed to understand the contexts and psychosocial factors that favor these adaptive transformations and allow the development of more evidence-based effective interventions for enhancing security in an attachment to God.

Similar processes have been found in studies examining attachment to a pet. First, there is extensive evidence that pet owners feel close to their pets, seek and enjoy this closeness, and view pets as providers of comfort and relief in times of need [120,121]. Second, research findings have supported the correspondence hypothesis, according to which people who are less secure in their close relationships are more likely to be insecure in their attachment to a pet [122]. Third, there is evidence that a pet can actually act as a safe haven and secure base [123]. Specifically, the actual or symbolic presence of a pet in times of need has been shown to reduce pet owners’ physiological signs of distress and to increase their confidence in goal pursuit. However, these effects were found only among pet owners who maintained a secure attachment to the pet [123]. Fourth, secure attachment to pets has been found to be positively associated with owners’ heightened emotion regulation and mental health [122]. 

Overall, these findings indicate that pets can sometimes serve as a safe haven and secure base and that secure attachment to a pet can benefit the pet owner’s mental health. As with attachment to God, the beneficial effects are greater for people who are more secure in close human relationships because they are more likely to form a secure attachment with a pet. For insecure pet owners, who tend to project their negative working models onto pets, the use of a pet as an attachment figure will fail to provide emotional compensation and be unlikely to repair attachment injuries. Nevertheless, this projection of negative working models onto a pet might be buffered during pet-assisted interventions aimed at helping owners to form a more secure attachment with their pets, thereby benefiting from the secure base a pet can provide [124]. We need more systematic research examining this possibility as well as the development of effective attachment-based pet-assisted interventions. 

Supportive group interactions can also bring about positive changes in group members’ attachment orientations and thereby contribute to their adjustment and health. According to Smith, Murphy, and Coats [125], a group can serve attachment functions by providing a safe haven and a secure base. That is, people can use a group as a symbolic source of support and safety in times of need and as a secure base for exploration and learning. As with other symbolic figures, however, appraising a group as a security provider can be distorted by group members’ attachment insecurities in close relationships, which may make it harder to form a secure attachment to groups. Indeed, Smith et al. [125] constructed a self-report scale to measure group-oriented attachment orientations and found that group-oriented insecurities were positively associated with attachment insecurities in close relationships. Moreover, Rom and Mikulincer [126] found that attachment insecurities in close relationships were associated with more negative appraisals of group interactions, lower self-efficacy in dealing with group tasks, more negative emotional reactions during group activities, more negative memories of group interactions, and worse actual performance in group missions (as assessed by both self-reports and observers’ ratings).

Although group attachment insecurities may be reflections or projections of interpersonal insecurities, positive experiences with groups might buffer the projection of previously established working models onto a particular group and favor the formation of a more secure group attachment. Following this reasoning, Rom and Mikulincer [126] examined the potential buffering effect of group cohesion, i.e., coordination, cooperation, support, and consensus among group members that facilitate learning and effective team performance [127,128]. From an attachment perspective, group cohesion refers to the extent to which group members feel protected, comforted, supported, and encouraged by the group. Findings indicated that group cohesion attenuated attachment-anxious people’s tendency to project their negative working models onto a group and improved their socioemotional and instrumental functioning during group missions. Conceptually similar findings were reported by Ames et al. [129], who showed that participation in a facilitation group before beginning college had beneficial effects on attachment-anxious participants’ academic and emotional adjustment six months later. Gallagher, Tasca, Ritchie, Balfour, and Bissada [130] also found that cohesiveness of a therapeutic group had beneficial effects on the mental health of attachment-anxious group members undergoing group psychodynamic therapy. 

These findings are in line with McCluskey’s [131] contention that “failures in early attachment relationships can be revisited within the context of therapeutic groups and that groups can provide the context for supporting authentic connection with one’s own effect and encourage resonance with the effect of other people” (p. 140). More research is needed, however, on the psychological and interpersonal processes through which cohesive groups might help insecure adults repair attachment injuries. Future research should also examine other group-level characteristics that might be critical for security enhancement (e.g., group size, group members’ personality heterogeneity). In addition, evidence is needed concerning the relative contribution to secure group attachment of security-enhancing interactions among individual group members and the cohesiveness of a group over and above the qualities of its individual members. This is a major task for future studies.

Social institutions (e.g., the judicial system, police, government) can also be viewed as potential providers of security. In the realm of politics, there are examples of citizens seeking safety, security, and support in a self-declared “stronger, wiser” political leader who promises to restore and sustain security [94]. In this context, Bar-Tal [132] eloquently described a type of political leader that deliberately exaggerates threats and dangers in order to arouse fear and insecurity, and then presents him- or herself as the society’s savior who will deliver safety and security. Similar attachment dynamics are evident in Stern’s [133] description of the ways in which people are recruited to become violent terrorists. Recruiters target highly insecure people and then bring them progressively into line with the aims of terrorist groups or religious cults by alternately reactivating their sense of insecurity (by means of humiliation and self-denigration) and then reducing it through praise and applause from the group and its leaders. In this way, followers can identify with the grandiosity of a destructively charismatic leader who promises security, safety, and permanent approval (martyrdom) to compensate for their sense of helplessness or meaninglessness.

In his recent book on religion and attachment, Granqvist [106] proposed that a public welfare system, by providing a material safety net in the event of adverse life events (e.g., public health, unemployment funding), can heighten citizen’s confidence in the availability and responsiveness of government in times of need and strengthen their feelings of being protected and cared for. Granqvist speculated that this kind of support is partly responsible for the strong negative correlation between a country’s social welfare benefits and its overall religiosity. The benefits that people once sought from attachment to God seem to be partly obtainable from secular social support systems. 

In the field of law, Blader and Tyler [134] claimed that a social institution high on procedural justice (i.e., whose procedures are perceived as fair and just) leads people to feel well-treated by the institution and to trust its goodwill. These feelings resemble the way a secure child feels toward a caring and loving parent. In fact, as a security-enhancing parent increases children’s compliance with rules and their cooperation with socialization processes [42], institutions that proceed in a fair and just way also increase people’s security, social cooperation, and engagement [134]. There is accumulating evidence that attachment security is associated with heightened trust in organizations [135], more ethical organizational decisions [136], and more pro-organizational altruistic behaviors (what Organ [137] called organizational citizenship behaviors), e.g., [138].

Similar issues arise in situations of intergroup conflict and violence; if the parties do not trust each other’s goodwill and fairness, and instead insist on viewing each other as mortal threats, they cannot attain stable peace. In all of these life domains, one sees the continual importance of security enhancement that can benefit both individual and societal well-being. Of course, these theoretical analogies should be empirically tested in order to understand the attachment-related implications of societal systems and ideologies and to identify the mechanisms by which social institutions can enhance security. 

## 12. Conclusions

In this article, we have briefly reviewed the theory and a sampling of research findings concerning the ways in which the sense of attachment security can be enhanced in the laboratory, within actual relational contexts, and through symbolic personages, groups, and societal institutions. We hope we have demonstrated the broad relevance of attachment theory and research to the domains of parenting, marital relationships, counseling and psychotherapy, education, health and medicine, leadership and management, groups and organizations, justice, and government. Noticing and nurturing the attachment aspects of all these domains could make an enormous contribution to individuals’ mental and physical health and quality of life. There is now adequate empirical knowledge to inspire future applications and interventions. Of course, the efficacy of these efforts will need to be evaluated by high-quality research, and this research may have a bearing on how broader attachment theory, or an integration of attachment theory with other related theories designed for various social domains, is to be attempted.

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
