# Peer review of "Enhancing the “Broaden and Build” Cycle of Attachment Security in Adulthood: From the Laboratory to Relational Contexts and Societal Systems"

_ijerph, 2020, doi:10.3390/ijerph17062054_

Round 1

Reviewer 1 Report

This manuscript focuses upon the attachment theory and collects findings of theory-application from different angels, perspectives and systems such as laboratory experiments, romantic and martial relationships, education, health and medicine, psychotherapy, public management and social welfare, social institutions etc.

This is an interesting paper, however, it lacks an empirical foundation and any kinds of data (either first-hand or second hand) to prove research findings. If the authors rather want to present a theoretical paper without reference to any empirical dataset, then the authors shall provide more in-depth theoretical analysis and attempt to hybridize, synthesize and integrate different theoretical approaches, models and concepts into a new created theoretical framework, and noticeably clarifying their new theoretical contributions to this field. Otherwise, without empirical basis, without deductive and inductive thinking process, without theoretical innovation, without formulating hypotheses and their examination, this article looks more like a literature review and summary of existing studies in academia.  

Author Response

The current manuscript is indeed a literature review and a summary of existing academic studies, and therefore we did not make any change in the structure and agenda of the paper.

Reviewer 2 Report

I greatly enjoyed reading this review, and overall I felt that it covered the relevant literature succinctly and thoroughly. My concerns are relatively minor, mostly with regard to theoretical tie-ins and clarification of reviewed findings in a few places. I have outlined them by page number below:

p. 2: At the end of the first paragraph, I believe it would be worth citing a few seminal papers that brought attachment research into adult romantic contexts (e.g., Hazan & Shaver, Collins & Read, etc.).

p. 2: I think readers would benefit from a little more explicit clarification regarding security priming. Are activated mental representations of security meant to shift one's context-specific or partner-specific working models, or to induce malleability in the top-node dominant attachment orientation? Or is a more apt interpretation that repeated experiences of priming to a specific working model could eventually lead to shifts in one's dominant orientation? 

p. 3: Broaden-and-build seems to be a fruitful framework for characterizing induced or naturally occurring changes to one's attachment security. I would, however, have appreciated more explicitly stated links between the two. For example, what are the specific broadened thought-action repertoires when security is enhanced? Are they a mixture of secure base scripts and expectations regarding safe haven support? Similarly, are the resources that are built largely social (via attachment figures), psychological (in terms of safety and security or resilience), or all of the above? The connections between attachment theory and the broaden-and-build model are elegant and implied by much of this review, but I feel that a few more explicit tie-ins between the two could more strongly make the case for why the review is couched in terms of broadening and building instead of some other candidate framework.

p. 3: Is there any evidence to suggest that individuals who are more chronically attachment avoidant vs. anxious respond more securely to differently designed security primes? Can the authors say a bit more about what all security primes share psychologically (e.g., secure base and/or safe haven functions)?

p. 4: I'd like to see a brief note regarding how attachment figures were nominated in the reviewed studies. Was a standard form used, or were individuals who fulfilled specific attachment functions targeted?

p. 7: The claim that "...attachment-based approaches are more effective than other cognitive-behavioral or psychodynamic approaches in improving mental health and psychosocial functioning" is a strong one. I see there are citations to support the argument, but given the vastness of the scope of clinical approaches, it could be valuable to add a sentence or two explaining the specific domains in which this has been found.

p. 7: For what age groups have these teacher-student attachment benefits been found?

p. 7: Are there limits to the benefit of the "stronger and wiser" attachment figure? Does this only apply if they are domain experts and there is some role hierarchy? For example, would such an attachment figure undermine egalitarian relations for a married couple?

p. 8: Are there psychometric indicators for the self-report scale assessing healthcare providers' safe haven and secure base functions?

p. 8: In terms of evidence-based physician training programs in secure base and safe haven functions, I wonder if there might be programs that accomplish similar goals (but without attachment-specific labels). Perhaps training programs in osteopathic medicine, for example.

p. 8: Is there any evidence speaking more directly to whether leaders who support attachment functions promote improved performance (in addition to morale, resilience, and satisfaction)? Perhaps "instrumental functioning" is analogous to performance, but certain leadership philosophies encourage high expectations and harsh feedback in the service of coaxing peak performance. Such approaches wouldn't necessarily lack safe haven and/or secure base provision, but I am curious whether there are boundary conditions for the positive effects of sensitivity and emotional support in certain competitive team settings.

p. 10: Is a particular aspect of attachment to a group especially critical for security enhancement? Is it about attachment to individual group members, to the group as an entity over and above its individual membership, and/or to its leadership?

p. 11: Given the paper's title, I was anticipating a lengthier discussion of the possible role of social systems in supporting attachment security. There was only really one paragraph addressing this, and the connections to attachment security seemed a bit loose (the limitation of the analogy was acknowledged by the authors). If some work on the role of attachment style in social participation or organizational settings could be brought to bear a little more directly on this category, perhaps it would feel like a stronger concluding segment to the review overall. 

Author Response

  1. We added citations for a few seminal papers that brought attachment research into adult romantic contexts at the end of the first paragraph of p. 2 (lines 56-57) and added these entries to the references list (entries 5, 6, and 7).
  2. On p. 2 lines 72-81, we explicitly noted that (a) activated mental representations can shift one's context-specific or partner-specific working models and induce some malleability in the top-node dominant attachment orientation, and (b) repeated activation of a specific working model could eventually lead to congruent shifts in one's dominant attachment orientation.
  3. We re-wrote the section entitled “The Broaden-and-Build Cycle of Attachment Security“ (pp. 2-3, lines 89-141) and explicitly note that (a) security enhancement fosters both positive working models of others and the self and a “secure-base script,” and (b) both interactions with responsive attachment figures (at the interpersonal level) and the activated positive working models and secure-base script (at the intrapersonal level) contribute to resilience and the broadening of perspectives and skills.
  4. On p. 5 lines 228-238, we added a paragraph noting that the psychological effects of explicit security priming tend to be more dependent on a participant's dispositional attachment orientation than the effects of implicit security priming. In the same paragraph, we also noted that the current state of the literature does not allow us to delineate the specific stimuli and manipulations that are the most effective security primes for enhancing security in either attachment-avoidant or attachment-anxious people.
  5. On p. 4 lines 180-183, we explicitly noted that the common denominator of all the security priming techniques is that the primed figure accomplishes both safe-haven and secure-base functions.
  6. On p. 4 lines 169-180, we described how attachment figures are nominated in the reviewed studies. In some of the studies, participants received the 6-item WHOTO scale and provided the name of a specific person who fits the role of a security provider as described in each of the items (proximity seeking, safe haven, secure base). In other studies, participants received instructions to think about a specific person who support and comfort them when distressed and they trust he or she will be available and responsive when needed. We also briefly described the nomination technique in some of the studies that we reviewed in this section (p.4 line 196, p. 5 lines 218 and 225).
  7. On p. 8 lines 369-370, we explicitly wrote that evidence concerning the effectiveness of attachment-based approaches (compared to other cognitive-behavioral or psychodynamic approaches) has been found among patients diagnosed with eating disorders, depression, and personality disorders.
  8. On p. 8 lines 381 and 384, we noted that the reviewed teacher-student attachment benefits have been found mainly among elementary-school children.
  9. On p. 6 lines 260-262, we explicitly noted that there are cases in which the potential security provider is a domain expert and occupies the role of a “stronger and wiser” caregiver in a formal role hierarchy (e.g., teacher, coach, therapist, manager). In the same paragraph, we also wrote that in egalitarian relations, such as friendships and romantic relationships, each partner can informally occupy the role of a “stronger and wiser” caregiver when the other partner is in need and asks for safe-haven or secure-base support (p. 6, lines 262-265). In addition, we delineated the limits of the benefit of an attachment hierarchy in egalitarian relationships. Specifically, when one partner chronically occupies the role of the needy care-seeker or monopolizes the role of the stronger caregiver in these egalitarian relations, it unbalance and potentiallly damages relationship quality and stability (p. 6, lines 265-267).
  10. On p. 9 lines 421-422 424-425, we wrote that (a) the scale used by Maunder and Hunter had adequate internal consistency, and (b) their research is only a preliminary study and more data should be collected on the psychometric properties of the scale.
  11. On p. 9 lines 443-450, we noted that, although not derived from attachment theory and research, several psychoeducation programs have been developed to enhance medical students and physicians' capacity to respond empathetically to patients. We also wrote that we are confident that construing these programs as means for enhancing patients' attachment security and helping medical students and physicians to soften patients' insecurities would aid healing and reduce expenses for both clients and the medical system.
  12. On p. 10 lines 467-470, we noted that Davidovitz et al. (2007) found that an officer’s ability to provide effective emotional and instrumental support to his soldiers in times of need was positively associated with his soldiers’ reports of socioemotional functioning and task performance in the military unit. In addition, we wrote that future studies should extend this line of research and examine the potential effects of managers’ responsiveness on actual task performance (p. 10, lines 481-482). We also noted that this is an important that because some organizations encourage high expectations and harsh feedback in the service of coaxing peak performance and then one should systematically examine whether managers’ responsiveness would have positive effects on task performance within these competitive and demanding organizational settings (p. 10, lines 482-486).
  13. On p. 12 lines 589-593, we wrote that (a) future research should examine other group-level characteristics that can be critical for security enhancement (e.g., group size, group member's personality heterogeneity), and (b) we don’t have any evidence about the relative contribution of security-enhancing interactions with individual group members and the cohesiveness of a group as an entity over and above its individual membership to the formation of secure group attachment. We strongly urge further research on this issue.
  14. In line with the reviewer's suggestion, we expanded our discussion of the possible role of security enhancement in the fields of politics and law and added relevant references concerning the involvement of attachment style in social participation and organizational settings (pp. 12 lines 594-606 and p. 13 lines 621-627).